# Current Value of Biparametric Prostate MRI with Machine-Learning or Deep-Learning in the Detection, Grading, and Characterization of Prostate Cancer: A Systematic Review

**DOI:** 10.3390/diagnostics12040799

**Published:** 2022-03-24

**Authors:** Henrik J. Michaely, Giacomo Aringhieri, Dania Cioni, Emanuele Neri

**Affiliations:** 1Medical Faculty Mannheim, University of Heidelberg, 69120 Heidelberg, Germany; 2Academic Radiology, Department of Translational Research, University of Pisa, 56126 Pisa, Italy; giacomo.aringhieri@unipi.it (G.A.); dania.cioni@unipi.it (D.C.); emanuele.neri@unipi.it (E.N.); 3Italian Society of Medical and Interventional Radiology, SIRM Foundation, Via della Signora 2, 20122 Milano, Italy

**Keywords:** prostate cancer, multiparametric prostate MRI, biparametric prostate MRI, deep-learning, radiomics, artificial intelligence, cancer detection, PIRADS

## Abstract

Prostate cancer detection with magnetic resonance imaging is based on a standardized MRI-protocol according to the PI-RADS guidelines including morphologic imaging, diffusion weighted imaging, and perfusion. To facilitate data acquisition and analysis the contrast-enhanced perfusion is often omitted resulting in a biparametric prostate MRI protocol. The intention of this review is to analyze the current value of biparametric prostate MRI in combination with methods of machine-learning and deep learning in the detection, grading, and characterization of prostate cancer; if available a direct comparison with human radiologist performance was performed. PubMed was systematically queried and 29 appropriate studies were identified and retrieved. The data show that detection of clinically significant prostate cancer and differentiation of prostate cancer from non-cancerous tissue using machine-learning and deep learning is feasible with promising results. Some techniques of machine-learning and deep-learning currently seem to be equally good as human radiologists in terms of classification of single lesion according to the PIRADS score.

## 1. Introduction

### 1.1. Prostate Cancer

Prostate cancer (PCA) is the second most common cancer in men worldwide and it accounts for up to 25% of all malignancies in Europe [1]. It is the third leading cause of cancer-related death in the United States and Europe [2,3]. The incidence of prostate cancer increases with rising age of patients, and prostate cancer and its management are becoming a major public health challenge. PCA aggressiveness can be linked to specific genes such as BRCA, and behavior such as smoking [4,5]. Accurate and early detection of prostate cancer is therefore paramount to achieve good overall patient outcomes. The tools available for assessing and detecting prostate cancer are digital rectal examination (DRE), PSA screening, transrectal ultrasound, and MRI whereby the latter received the highest amount of attention in the past decade due to its unprecedented capabilities in accuracy [6,7,8].

In contrast to ultrasound and digital rectal examination, MRI offers an operator-independent tool for objectively assessing the entire prostate gland from base to apex and from the posterior peripheral zone (PZ) to the anterior fibromuscular stroma (AFMS) that are barely assessable with DRE [6,9].

Magnetic resonance imaging of the prostate has a long history going back more than 20 years. In the initial phase, high resolution T2-weighted (T2w) imaging and spectroscopy were mainly used as tools for detecting prostate cancer. Yet, spectroscopy is slow and susceptible to artefacts and was not well perceived. In the recent decade, further developments have taken over including diffusion weighted imaging (DWI), dynamic contrast enhanced imaging (DCE). The entire prostate exam has been standardized worldwide and its reporting has been harmonized by the PIRADS (Prostate Imaging Reporting and Data System) system [10]. This classification system allows to objectively assess the prostate and potential cancerous zones and standardizes reporting over separate sites so that the overall performance of MRI is increased and is more reproducible compared to previous periods. With this development MRI of the prostate follows the trend to standardize the entire radiological procedure from image acquisition to data reporting to achieve a higher reliability, enhanced reproducibility, and a direct implication for radiology-based treatments as it has previously successfully demonstrated in breast imaging with BIRADS (Breast Imaging Reporting and Data System) [11].

The report structuring provided by PIRADS is already a condensation of the imaging information and standardizes reporting and its output. This is one major step toward a more automated and operator-independent radiology. Moreover, the image acquisition parameters, slice orientations, and sequences with its specific sequence characteristics are governed by PIRADS [12]. This automatically sets the stage for a potential automated image analysis. In the past decade, artificial intelligence (AI) with its subdivisions of machine learning (ML), radiomics, and deep learning (DL) has become more prevalent. At this point in time, ML and DL are still no clinical standards. Radiomics, for example, use quantitative imaging features that are often unrecognizable to the human eye. Therefore, it is increasing the number of potential parameters to the multi-parametric approach of prostate MRI and with potential benefits for PCA detection and grading and beyond. DL techniques such as convoluted neural networks (CNN) are currently considered gold standard in computer vision and pattern recognition and hence have potential benefits for PCA detection and grading. With larger data sets as basis, they have the potential to automatically learn and deduct conclusions so that PCA recognition based on unperceivable features to the human eye might be possible. Despite numerous experimental studies which will be discussed further in this study, there is no standardized approach on how to use and implement DL and ML for prostate imaging now.

The aim of this study is to elucidate the status of artificial intelligence in prostate imaging with a focus on the so-called bi-parametric (bp) approach of prostate MRI (bpMRI).

### 1.2. Prostate Imaging Reporting and Data System

PIRADS was established by key global experts in the field of prostate imaging from America and Europe (European Society of Urogenital Radiology (ESUR), American College of Radiology (ACR)) to facilitate and standardize prostate MRI with the aim of assessing the risk of clinically significant prostate cancer (csPCA). The first version of the PIRADS recommendations was published in December 2011, the latest and current update was published in 2019 (PIRADS v2.1) [10,12,13].

Various studies have compared the predictive performance of PI-RADS v1 for the detection of csPCA compared to image-guided biopsy or radical prostatectomy (RP) specimens as standard of reference. In a 2015 study, Thompson reported multi-parametric MRI detection of csPCA had sensitivity of 96%, specificity of 36%, negative predictive value and positive predictive values of 92% and 52%; when PI-RADS was incorporated into a multivariate analysis (PSA, digital rectal exam, prostate volume, patient age) the area under the curve (AUC) improved from 0.776 to 0.879, *p* < 0.001 [14]. A similar paper showed that PI-RADS v2 correctly identified 94–95% of prostate cancer foci ≥ 0.5 mL but was limited for the assessment of Gleason Score (GS) ≥ 4 + 3 csPCA ≤ 0.5 mL [15]. An experienced radiologist using PIRADS v2 is reported to achieve an AUC of 0.83 with 77% sensitivity and 81% specificity [16].

### 1.3. Sequences for Prostate MRI

The initial protocol for MRI of the prostate as provided by PIRADS included high-resolution multiplanar T2w-imaging, DWI, and DCE after the intravenous administration of paramagnetic gadolinium chelate contrast agent. This so-called multi parametric prostate MRI (mpMRI) is considered as the gold standard. T2w-imaging is used to demonstrate zonal anatomy of the prostate. Tumors can be well delineated, and their relation to the prostate capsule can be examined. Benign changes such as benign prostate hyperplasia, post-prostatic changes of the peripheral zone or scars can be identified. T2w-imaging is considered the gold standard for the transitional zone (TZ) of the prostate gland. In addition, T2w-imaging can be used to measure the volume of the prostate. The high anatomic information content of T2w-imaging makes this sequence the perfect roadmap for image-guided biopsy [12,17].

DWI serves as an indirect measure of cellular density. In case of a malignant tumor with high cellular density, the ability of water to freely move in the interstitial compartment is decreased hence the diffusion is impaired. The images with high b-values and even those with more and more common-interpolated calculated b-values allow quick and easy depiction of these suspicious areas in the prostate. The calculated ADC maps give a quantitative measure of cellular density and can be considered as a molecular imaging tool for tumor aggressiveness. DWI imaging is considered as the reference sequence for the peripheral zone (PZ) of the prostate [12,17].

Dynamic contrast enhancement (DCE) is considered as the weakest of the three used approaches for prostate imaging. In contrast to T2w-imaging and DWI, DCE is not being considered as a dominant sequence for any of the prostate zones. It only serves as a tiebreaker in very specific questions in the PIRADS system. In addition, it requires the intravenous administration of contrast agent with the risk of side-effects such as allergies, nephrogenic systemic fibrosis, or Gadolinium deposition in the body [18,19,20,21]. While the risk of nephrogenic systemic fibrosis is controllable by using little amounts of macrocyclic Gd-chelates, no harmful consequence for Gd-chelate depositions in the body has been found [22,23]. Nevertheless, patients often try to avoid contrast agent if feasible. Moreover, physicians embrace the idea of non-enhanced exams equally, as it speeds up the acquisition and reduces the number of potential complications. In addition, omitting contrast agent permits to save money.

### 1.4. Multiparametric and Biparametric MRI of the Prostate

With this in mind and the knowledge that the performance of DCE often yielded limited added value to T2w-imaging and DWI in mpMRI of the prostate bi-parametric MRI (bpMRI) of the prostate is gaining considerable support [15]. Meanwhile, there are several high-ranked studies such as the PROMIS trial and meta-analyses comparing mpMRI and bpMRI of the prostate [24,25,26]. Current data underline the high negative value of bpMRI in biopsy-naïve patients with a negative predictive value of up to 97% [27,28]. Whether bpMRI might be slightly less accurate in less-experience readers is not yet clearly proven [29,30]. A currently accepted position is that bpMRI of the prostate seems to be equally good as mpMRI of the prostate for patients with low and high risk for csPCA but DCE might be of worth in patients with intermediate risk and PIRADS 3 lesions [25,26,31,32,33,34,35] (Figure 1). bpMRI of the prostate is also commonly used for computer-based postprocessing using artificial intelligence. This is due to the fact that DCE contains a fourth dimension (time) which make those images harder to algin and match with two-dimensional anatomical images such as T2w-imaging and DWI. Another drawback of DCE is that image information is not obvious. The image information on contrast media arrival and distribution which is seen as a surrogate marker for microvascular density have to be extracted using semiquantitative or quantitative pharmacokinetic models which adds another layer of complexity on postprocessing, along with the increase of time necessary to report the exams.

### 1.5. Artificial Intelligence (AI) for Image Postprocessing

The availability of cheap and high computing power with the additional advent of postprocessing technologies and artificial intelligence such as machine learning techniques and deep neural networks has fostered the application of those techniques for radiology tasks such as tumor detection. The current hierarchical concept of AI is depicted in Figure 2.

*Machine-learning* (ML) is a subfield of AI in which algorithms are trained to perform tasks by learning rules from data rather than explicit programming. *Radiomics* is seen as a method that extracts large numbers of features from radiological images using data characterization algorithms such as first order statistics, shape-based, histogram-based analyses, Gray Level Co-occurrence Matrix, Gray Level Run Length Matrix, Gray Level Size Zone Matrix, Gray Level Dependence Matrix, Neighboring Gray Tone Difference Matrix to name a few [36,37,38,39]. These features are said to have the potential to uncover disease characteristics that are hard to be appreciated by the naked eye. The hypothesis of radiomics is that distinctive imaging features between disease forms may be useful for detecting changes and potentially predicting prognosis and therapeutic response for various conditions such as e.g., detection of csPCA. These radiomic features are then often further analyzed using ML-techniques. An example of a radiomics ML-workflow is shown in Figure 3. An issue concerning ML-techniques is that it often requires the manual placement of a region of interest hence hereby introducing a potential source for errors and biases.

*Deep learning* (DL) is a subfield of AI in which algorithms are trained to perform tasks by learning patterns from data rather than explicit programming. The key factors for the increasing attention that DL attracted in the past years are the availability of large quantities of labelled data, the inexpensive and powerful computing hardware particularly graphic-processing units and improvements in training techniques and architectures. DL is a type of representation learning in which the algorithms learn a composition of features that reflect the hierarchy of structures in the data. Current state-of-the-art for medical image recognition using DL techniques are so called convoluted neural networks (CNN). These networks are characterized by an architecture of connected non-linear functions that learn multiple levels of representations of the input data thereby extracting possibly millions of features [41]. Especially CNNs in which a series of convolution of filter layers are exploited are suitable for image processing [42]. Newer techniques such as transfer learning and data augmentation, or the application of generative methods help in mitigating existing limitations of CNN [43]. The entire process of data processing within the multiple layers of a CNN with convolution filters, pooling, and maximum filtering is beyond the scope of this study. Largely simplified, one might say that bottom layers of the CNN act as a feature extractor while the top layers of the CNN act as a classifier. An overview is given in Figure 4 in which the DL workflow is compared to radiomics or the standard radiology reading process [44]. The reason that CNN-based approaches are considered superior to radiomics is that radiomics depend on hand-crafted features which is limited, whereas CNN can generate features that are most appropriate to the problem itself [45].

## 2. Materials and Methods

Literature research for this study took place in August 2021. A PubMed query with the search terms “prostate” and “magnetic” and “deep learning” or “machine learning” or “radiomics” was performed. The aim was to retrieve those studies which made use of ML or DL techniques to facilitate tumor detection and grading. To make sure that only current techniques were included in the analysis only publications from the year 2019 to 2021 were included. Particularly in the field of CNN the technical improvement is rapidly evolving so that elder publications might not represent the current state-of-the-art. Total of 95 publications were initially retrieved. Of these, 66 were omitted for several reasons so that 29 publications were available for analysis (see Figure 5). Clinical data (question to be answered, number of patients, age, AI-technique, lesion segmentation, MRI-technique, sensitivity, specificity, accuracy, AUC) were then manually extracted and transferred to a Microsoft Excel 365 spreadsheet (Microsoft, Redmond, WA, USA). PRISMA guidelines were followed [46]. An overview of the study according to the PRISMA guidelines can be found in the Appendix A.

This paper focuses on bpMRI. The current PIRADS guidelines state: “Given the limited role of DCE, there is growing interest in performing prostate MRI without DCE, a procedure termed “biparametric MRI” (bpMRI). A number of studies have reported data that supports the value of bpMRI for detection of csPCA in biopsy-naïve men and those with a prior negative biopsy”. The potential benefits of bpMRI include: (1) elimination of adverse events and gadolinium, (2) faster MRI-exam times, and (3) overall reduced costs [47]. These factors will potentially make bpMRI easily accessible. Remaining concerns are that the DCE sequence may serve as backup in case of image degradation of the DWI or T2w sequence. It seems as if DCE may be of less value for assessment of treatment of naïve prostate patients but remains essential in assessment for local recurrence following prior treatment, which however is a setting in which current PI-RADS assessment criteria do not apply. The conclusion of the PIRADS steering committee therefore advocates the use of mpMRI particularly in (1) patients with prior negative biopsies with unexplained raised PSA values, (2) those in active surveillance who are being evaluated for fast PSA doubling times or changing clinical/pathologic status, (3) men who previously had undergone a bpMRI exam that did not show findings suspicious for csPCA, and who remain at persistent suspicion of harboring disease, (4) biopsy-naïve men with strong family history, known genetic predispositions, elevated urinary genomic scores, and higher than average risk calculator scores for csPCA, and (5) men with a hip implant or other consideration that will likely degrade DWI [47].

For this paper bpMRI was selected as most studies dealing with ML or DL techniques solely relay on T2w-imaging and DWI. DCE data were rarely included. In contrast to T2w-imaging and DWI the DCE-data must be postprocessed first to generate parameter maps. This process is not yet standardized as several pharmacokinetic models and hereof derived software implementations for postprocessing exist. Without generation of parameter maps a huge number of images would have to be fed into the ML/DL algorithms—a step that most research groups obviously did not want to undertake.

## 3. Results

All included studies are listed with an abbreviated overview in Table 1.

Total of 29 studies were included in this study. Thirteen of them used ML (44.8%), 14 of them used DL-techniques (48.2%), and 2 of them used a combination of ML and DL (6.9%). The data for 27 of the studies were acquired at 3T (93.1%), 2 of them were acquired at 1.5 T (6.9%). A total of 7466 patients were analyzed within this data set. Hereby, the ProstatEx-data set from the Radbound University, The Netherlands was used seven times. The smallest study had a sample size of 25 patients, the largest study had a sample size of 834 patients. The MRI-technique used for AI-postprocessing most often was T2w-imaging in combination with ADC map and DWI (15 studies/53.6%). Runner-up were T2w-imaging and ADC map (8 studies, 28.6%) and T2w-imaging and DWI (2 studies, 7.1%).

### 3.1. Tumor Detection and Grading

As seen in Table 1, the results (AUC, sensitivities and specificities) were comparable and no trend clearly favoring ML or DL-approaches in terms of superiority could be detected. Most studies required manual interaction in which a radiologist had to segment the region of interest.

Overall, the rate of detection and correct tumor creating using AI-techniques was comparable to the performance of trained radiologists in most studies. Studies were often hard to compare as they differed in terms of standard of reference (e.g., Gleason score (GS) vs. PIRADS vs. National Comprehensive Cancer Network Guidelines vs. ISUP Guidelines) and different cut-off values within the same grading system (e.g., GS 7 was in one study considered intermediate grade, in most studies considered high-grade tumor). Some studies focused on the PZ only, while others accepted the entire gland as target tissue.

In a small study with 33 patients to predict IMRT response, GS prediction and PCA stage, GS prediction using T2w-radiomic models was found more predictive (mean AUC 0.739) rather than ADC models (mean AUC 0.70), while for stage prediction, ADC models had higher prediction performance (mean AUC 0.675). For T2w-radiomic models, mean AUC was obtained as 0.625 [40].

Using T2w-imaging and 12 b-values from diffusion along with Kurtosis analysis and T2 mapping for differentiation GS ≤ 3 + 3 vs. GS > 3 + 3 an AUC of 0.88 (95% CI 0.82–0.95) could be reached. This study with 72 patients was the only one to employ T2 mapping which, after all, was deemed as of little worth [51].

In a stringent ML-Radiomics study, an equally high AUC for tumor grading according to National Comprehensive Cancer Network guidelines in low-risk vs. high-risk (i.e., GS ≥ 8) was found for the PIRADS assessment as well as for the ML-approach (0.73 vs. 0.71, *p* > 0.05) [49]. Interestingly, the precision and recall were higher with the ML-approach compared to the PIRADS assessment (0.57 and 0.86 vs. 0.45 and 0.61). Similar results were found for the discrimination of ciPCA and csPCA of the PZA using a ML-Radiomics approach with extreme gradient boosting [62]. In this study performed on the ProstatEx dataset, an AUC of 0.816 for the detection of csPCA using bpMRI was found. Adding DCE slightly increased AUC to 0.870, though this was not statistically significant. Based on the same data set but using optimized CNNs Zong et al. [63] concluded that adding ktrans from DCE deteriorated sensitivity and specificity when compared to bpMRI alone from 100%/83% to 71%/88%. The optimal reported AUC of this study was 0.84.

Extremely good ML-radiomics results for differentiation ciPCA vs. csPCA with an AUC of 0.999 were found in a study by Chen et al. They could also show that ML-radiomics exhibited a higher efficacy in differentiation ciPCA from csPCA than PIRADS. A potential explanation for this, compared to the other studies, is that outstanding result might be the study inclusion/exclusion criteria: small lesions <5 mm and lesion not well delineable on MRI were excluded [52].

Somewhat poorer results were presented in a study by Gong et al. [61]. Their ML-radiomics approach that was built on T2w-imaging and b800-DWI images yielded an AUC of 0.787 and an accuracy of 69.9% for the discrimination between ciPCA and csPCA. Adding clinical data to the MRI-based data slightly degraded the results with an AUC of 0.780 and an accuracy of 68.1%. A potential reason for this poorer outcome might be a different set of inclusion parameters.

Zhong et al. compared the performance of DL and Deep Transfer Learning (DTL) with experienced radiologists. They found that DTL further improves DL. The DTL results were comparable to radiologist’s performance using PIRADS v2. They concluded that DTL might serve as an adjunct technique to support non-experienced radiologists [54]. Similar results found a study using a CNN-trained algorithm to automatically attribute PIRADS scores to suspicions lesions. A performance comparable to a human radiologist was described [64]. The lowest agreement was found with low PIRADS score, getting better with higher PIRADS scores. There was no statistically significant difference between the radiologist-assigned PIRADS score and the AI-assigned PIRADS score with regards to the presence of csPCA for PIRADS 3–5.

In contrast, for Gleason score prediction one study found better results for AI-based approaches than radiologists for PZ and TZ [76]. This could be particularly useful in the context of active surveillance.

A different study looking into aggressiveness prediction (GS > 8) found equal AUCs for AI and radiologists but higher precision and recall rates for AI than PIRADS mitigating the problem of inter-reader variability [49].

An uncommon approach was presented in [65]. The authors hereby combined Radiomics and DL-based on bpMRI with DCE and T2w-imaging. No ADC/DWI-images were used. In few patients they included, promising results with an AUC of 0.96–0.98 for Gleason score prediction were found. No further study used this subset of DCE and T2w-imaging.

The prospective IMPROD trial also examined if the addition of clinical data and RNA expression profiles of genes associated with prostate cancer increased the accuracy for detection of csPCA [58]. In this study the bpMRI based data yielded the highest AUC 0.92. Adding RNA-based data or clinical data neither improved the results nor yielded better results by itself.

Cao et al. developed an FocalNet to automatically detect and grade PCA (Figure 6) [72]. A similar work was presented by Schelb et al. [75] where a U-Net was trained to detect, segment, and grade PCA. In comparison with radiologists’ PIRADS assessment, the U-Net sensitivities and specificities for detection of PCA at different sensitivity levels (PIRADS ≥3 and PIRADS ≥ 4) were comparable.

Positive results for DL-based techniques with a larger number of patients (*n* = 312) were found in a DL-Study by Schelb et al. using a U-Net [57]. They reported a sensitivity/specificity for radiologists using PIRADS for detection of PIRADS lesions ≥ 3 and 4 respectively of 96%/88% and 22%/50% while the U-Net approach yielded 96%/92% and 31%/47% (*p* > 0.05). In their study the U-Net also autocontoured the prostate and the lesion with dice-coefficient of 0.89 (very good) and 0.35 (moderate) respectively.

A ML-approach to generate “attention boxes” for the detection of csPCA was published by Mehralivand et al. [60]. Their multicentric approach with data from five institutions showed an AUC of 0.749 for PIRADS assessment of csPCA and a statistically non-significant AUC of 0.775 for the ML-based approach. For the TZ only, the ML-approach yielded a higher sensitivity for detection of csPCA than PIRADS (61.8% vs. 50.8%, *p* = 0.001). Interestingly, the reading time for the ML-approach was on average 40s longer.

An uncommon approach for CNNs was published by Chen et al. [66]. They used U-Net CNNs to segment the prostate and intraprostatic lesions hereby segmenting the PZ, TZ, CZ, and AFMS. Their approach demonstrated impressive results: a Dice coefficient of 63% and a sensitivity and specificity of 74.1% and 99.9% respectively for correctly segmenting the prostatic zones and the suspicious lesion. Yet, in contrast to most other studies, no grading or discrimination of the suspected PCA lesion was performed. As a segmentation study this study was included in this review as it included segmentation of the prostate and detection of the tumor within the prostate.

In a screening study with 3T-bpMRI, Winkel et al. [67] could include and analyze 48 patients, all above 45 years. In a biopsy-correlated reading two human readers and a commercial prototype DL-algorithm were compared in terms of detection of tumor-suspicious lesions and grading according to PIRADS. The DL-approach had a sensitivity and specificity of 87% and 50%. Noteworthy, the DL-analysis required just 14 s.

Different ML-based models were tested and found to be highly accurate for the diagnosis of TZ PCA (sensitivity/specificity/AUC): 93.2%/98.4%/0.989) and their discrimination from BPH-nodules. Reproducibility of segmentation was excellent (DSC 0.84 tumors and 0.87 BPH). Subgroup analyses of TZ PCA vs. stromal BPH (AUC = 0.976) and in <15 mm lesions (AUC = 0.990) remained highly accurate [48].

DL-approach for detection of csPCA in patients under active surveillance was brought up by Arif et al. [68]. Initially 366 patients with low risk were included of which 292 were included in the final study. Sensitivities and specificities for csPCA segmentation rose with increasing tumor volume: tumor volumes > 0.03 cc sensitivity 82% 7 specificity of 43%, AUC 0.65; tumor volume > 0.1 cc sensitivity 85%, specificity of 52%, AUC 0.73. Tumor volumes > 0.5 sensitivity 94%, specificity 74%, AUC 0.89.

A total of six studies among the included studies compared DL/ML-approach to human radiologists [52,57,60,64,72,75]. Overall, due to the small number of studies and because of the different approaches the results cannot be analyzed together. What these studies had in common however was the finding, that at this point AI-based methods revealed a performance similar to that of the radiologists’. No study could either show an advantage of AI-methods of the radiologists or vice versa. An overview about the results can be seen in Table 2.

### 3.2. PIRADS 3 Lesions

Radiomics can detect with high accuracy csPCA in PI-RADS 3 lesions [59,77]. Hou et al. examined in a ML-Radiomics approach the ability of bpMRI to identify csPCA in PIRADS 3 lesions in a group of 253 patients with PIRADS 3 lesions in the TZ and PZ of whom 59 (22.4%) had csPCA [59]. The ML-Radiomics approach including T2w imaging, DWI and ADC had an AUC of 0.89 (95% CI 0.88–0.90) for predicting the presence of csPCA in a PIRADS 3 lesion.

### 3.3. Extracapsular Extension and Biochemical Recurrence

He et al. set up a large study including 459 patients who underwent 3T bpMRI before prostate biopsy and/or prostatectomy [69]. The aim of the study was first to differentiate between benign and malignant tissue second to predict extracapsular extension (ECE) of prostate tumor and third to predict positive surgical margins (PSM) after RP. Using Radiomics they developed and tested an algorithm that was able to achieve an AUC of 0.905 for the determination of benign and malignant tissue, 0.728 for the prediction of ECE, and a 0.766 for the prediction of PSM. Similarly, Hout et al. found an identical AUC of 0.728 for the prediction of ECE in a DL-based approach using different CNN-architectures [73]. Hence one can infer from the information derived from prostate imaging not only the current situation in the gland but can also predict future developments that might take place under therapy.

Biochemical recurrence (BCR) prediction based on radiomics features was examined in T2w-images only with higher prediction of BCR (C-index 0.802) than conventional scores, particularly also higher than the Gleason scoring system (C-index 0.583) [74]. This work is of particular interest as it first, was one of the few multicentric studies (three centers) with a relatively large number of patients (485) and second, demonstrated the ability of DL-based CNN to look beyond the prostate and infer predictions on the future course of the disease/patient.

## 4. Discussion

Prostate cancer is a growing medical condition already now being the second most common cancer in men in the western world. The detection and grading of prostate cancer are shifting more toward MRI and is demanding a higher number of MRI-studies to be performed and read. Currently, prostate MRI is considered a specialized exam and requires a highly specific experience to be performed and reported with high quality. A first step toward facilitation of mpMRI prostate acquisition, reading, and reporting was PIRADS, but surely not the last step [10,12,13]. To put it in a nutshell: prostate MRI is developing from the holy grail, and only a few radiologists were being able to read it competently to a commodity in radiology. This is one of the key drivers behind the growing demand for computer-assisted diagnostic tools, such as tumor detection and grading, to facilitate the diagnostic interpretation of prostate MRI also for less-trained radiologists. As the prostate is a densely packed organ with much more information for example as the sparsely packed lung, simple machine learning tools based on e.g., density differences cannot be successfully employed. To distinguish the different prostatic tissues, such as normal transitional and peripheral zones and malignant tissue, higher-developed machine learning tools are required, often based on radiomics or even deep learning techniques. In the papers included in this review, most approaches using either ML or DL were similar to radiologists in their performance [49,54,57,64,75]. For some specific applications, such as tumor detection in the TZ or detection of clinically significant cancers in PIRADS 3 lesions, AI-based methods might even be superior to radiologists’ performance [48,59].

These AI-based approaches should enable less well-trained radiologists to read and report prostate-MRI reports with good quality [57,75]. The literature review showed that different approaches to tumor grading and characterization either via ML or DL are capable of differentiating between cancerous and non-cancerous tissue. New approaches are even able to autonomously segment the prostate and the tumor within the gland overcoming a limitation of the elder approaches, where radiologists often had to manually segment the lesions, resulting in a highly time-consuming task [72,75]. Apart from many site-specific implementations of radiomics, ML and DL, another sign of maturation of AI-based approaches is that a first commercial tool was already presented [67]. Compared to the other algorithms, this commercial tool was trained by big data sets for the initial training. This development underlines again the trend in imaging toward commoditization of imaging and democratization of information technology enabling every radiologist to perform on a high-quality imaging.

Yet, there are some obstacles still to overcome. First, MRI is a tricky imaging tool. A major drawback of MRI is the lack of standard quantification of image intensities. Within the same image, the intensities for the same material vary as they are affected by bias field distortions and imaging acquisition parameters, not always perfectly standardized. In addition, not only do MR images taken on different scanner vary in image intensities, but the images for the same patient on the same scanner at different times may appear differently from each other due to a variety of scanner- and patient-dependent variables [45]. Therefore, the initial step in ML/DL image postprocessing is to normalize the MR intensity [45]. This process could induce errors, however. At last, also the reproducibility of CNNs varies resulting in interscan differences, though with less impact [78]. Second, most studies rely on single site source data. Multicentric studies are very rare hence making it harder to compare results of AI-based algorithms across different vendors and sequence parameters. Third, the choice of imaging sequences and their specific parameters is variable. This work focused on bpMRI of the prostate. Even though for a radiologist imaging with T2w-imaging and DWI imaging would be seen as biparametric, things look different in the world of AI-based post-postprocessing: sometimes T2w and ADC, sometimes T2w and a single high b-value, T2w and multiple b-values or T2, ADC and b-values were used (hereby neglecting uncommon outlier studies using DCE and T2 or T1 and T2). Even though DWI source date and ADC are based on the same acquisition, their information content seems different. It was observed in one study that the use of CHB-DWI led to higher specificity while the use of ADC led to highest sensitivity, making the choice of sensing modality useful for different clinical scenarios [79]. For example, maximizing specificity is important for surgery for removal of prostate where minimizing false positive rates to avoid unnecessary surgeries is required. On the other hand, for cancer screening, maximizing sensitivity may be useful to avoid missing cancerous patients [79]. A clear definition what would be considered as truly bpMRI or standards for AI-postprocessing has not been set up. Yet, there is a first European initiative on the development and standardization of AI-based tools for prostate MRI [44]. Fourth, DL-based CNNs are notorious for being a “black box” in terms of the how the decision was achieved. While this may not be entirely true—CNNs can be monitored at any level at some expense—they might never be as transparent as ML-based approaches hence scaring some physicians from using them on real patients outside studies. Moreover, here, commercialization of the techniques might be helpful as larger companies have the means and money to certify algorithms with the FDA or the EU and thus make them broadly (commercially) available.

As seven studies made use of the ProstatEx data, it is worth looking at the overall conclusions the creators of the dataset and initiators of the contest published [80]: the majority of the 71 methods submitted to the challenge (classifying prostate lesions as clinically significant or not) the majority of those methods outperformed random guessing. They conclude that automated classification of clinically significant cancer seems feasible. While in the second contest (computationally assigning lesions to a Gleason grade group) only two out 43 methods did marginally better than random guessing. The creators also conclude that more images and larger data sets with better annotations might be necessary to draw significant conclusions, which brings up again the question of means and money. Another conclusion that can be drawn when looking at the included studies is that 3 T imaging seems to be the standard. This is partly because there is substantial overlap in the source data (ProstatEx) and that, of course, studies are being conducted at University Medical Centers which most often have state-of-the-art equipment. For radiology departments in smaller hospitals or private practices having a 3 T system is less likely. Regarding how far the results of 3T e.g., DWI can be transferred to 1.5 T and how the technological improvement of 1.5 T in the field of signal reception and processing is supportive remain unclear. One might speculate that a state-of-the-art 1.5 T will yield comparable image quality to an elder 3 T system. Looking at the source data of the different studies one can roughly estimate that 30% of these were acquired on elder (>14 a) 3 T systems.

There are some unexpected studies with novel approaches to patient care that should be to highlighted. One was therapy assessment with pre- and post-IMRT T2w-imaging [40] for “delta radiomics”, using radiomic features extracted from MR images for predicting response in prostate cancer patients. While there was only one study with this specific design, extrapolating ECE or BCR has roughly the same line of thought: could not it be possible to predict for changes in the future with imaging features measured today [69,73,74]. The AUC values of these studies were unexpectedly high (0.801–0.905) as well as the number of included patients.

### Limitations

This review has several limitations that need to be mentioned. First, ML and DL are extremely fast evolving techniques. Data provided in this review simply display a snapshot of the ongoing development. With the ever more powerful hardware and algorithms, future improvements seem likely. Most results are based on small feasibility studies, and larger applications of ML and DL in prostate imaging are not yet available. Whether their results match the promising initial studies remains unclear. Second, the inclusion criteria were narrow so that only 29 studies could be included. With the small sample size, different targets, and the different foci of the studies no wholistic analysis could be performed. Opening up the time window for the included studies would have led to inclusion of elder techniques potentially biasing the results.

## 5. Conclusions

In summary, this study investigated the current status of bpMRI of the prostate with postprocessing using ML and DL with a focus and tumor detection and grading. The presented results are very promising in terms of detection of csPCA and differentiation of prostate cancer from non-cancerous tissue. ML and DL seem to be equally good in terms of classification of single lesion according to the PIRADS score. Most approaches however rely on human interference and contouring the lesions. Only a few newer approaches automatically segment the entire gland and lesions, along with lesion grading according to PIRADS. There still exist a large variability and methods and just a few multicentric studies. No AI-postprocessing technique is considered gold standard at this time while there seems to be a trend toward CNNs. Regarding the actual MRI-sequences, most studies used T2w-imaging and either b-values from DWI or the ADC maps from DWI. The application of ML and DL to bpMRI postprocessing and the assistance in the reading process surely represent a step into the future of radiology. Currently however, these techniques remain at an experimental level and are not yet ready or available for a broader clinical application.

## Figures and Tables

**Figure 1 diagnostics-12-00799-f001:**
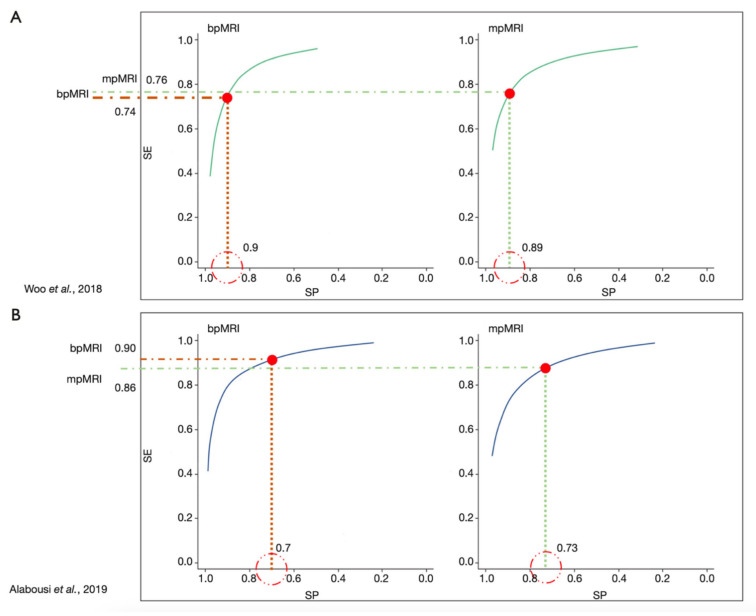
Overview of the performance of mpMRI and bpMRI based on data from Woo et al. [33] and Alabousi et al. [25] demonstrating the near equal performance of bpMRI to mpMRI (reprinted with permission from [17], Copyright 2020 Gland Surgery).

**Figure 2 diagnostics-12-00799-f002:**
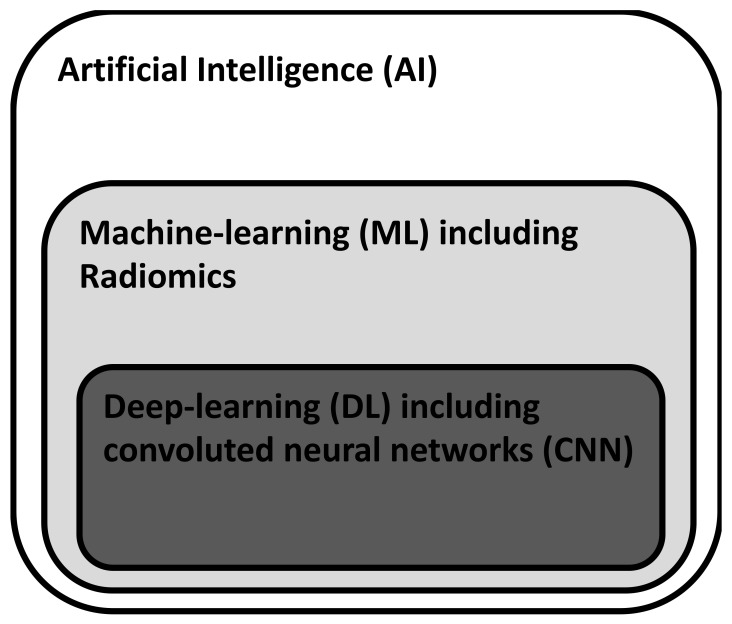
Hierarchical structure of AI-techniques. Whereas ML requires human feature engineering as guidance for learning, DL is based on self-learning algorithms that can detect and process simple and complex image features.

**Figure 3 diagnostics-12-00799-f003:**
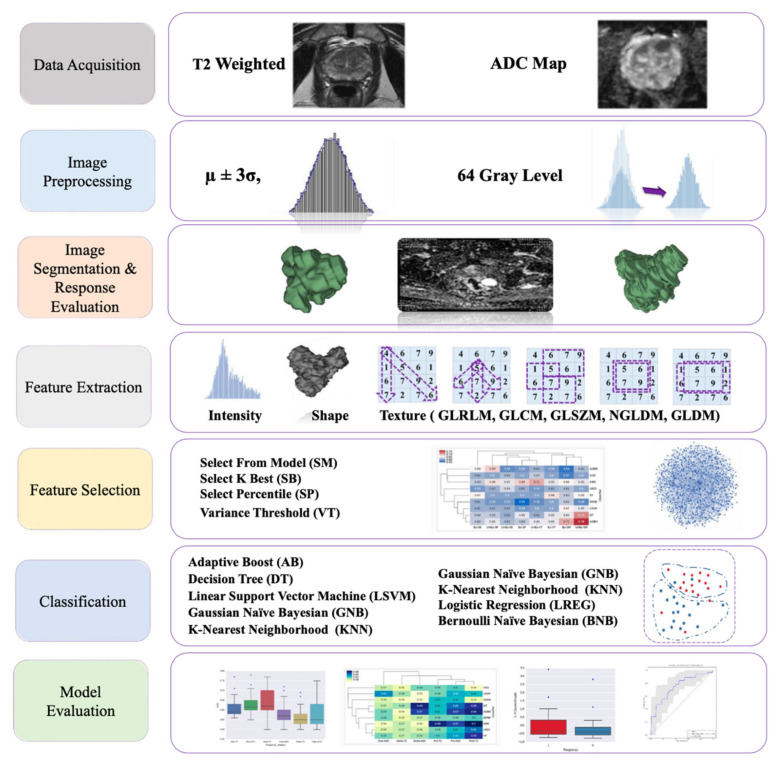
Sample radiomics workflow (reprinted with permission from [40], Copyright 2019 Springer Nature).

**Figure 4 diagnostics-12-00799-f004:**
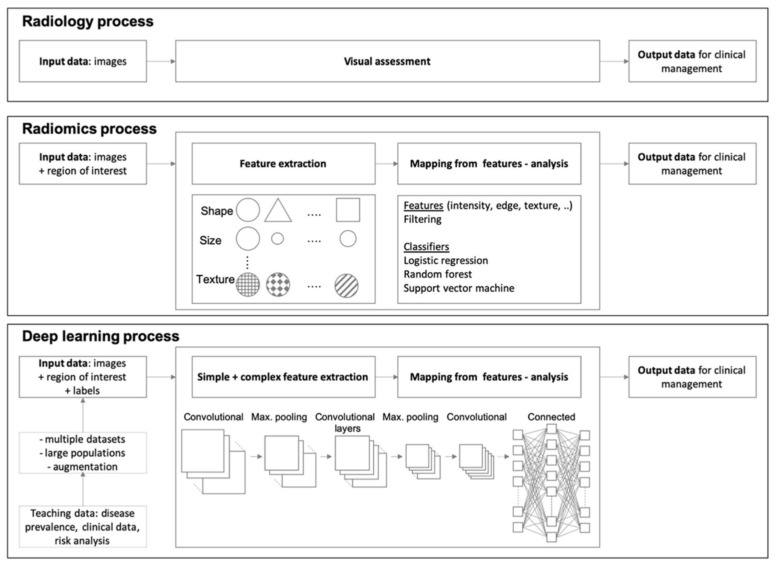
Workflow of standard radiology reporting compared to AI-based methods of radiomic and DL. The entire complexity of deep learning is only schematically shown. There is an abundance of different network architectures or CNN which are beyond the scope of this study. This figure only demonstrates a schematic CNN (reprinted under common creative license 4.0 from [44], Copyright 2021 Springer Nature).

**Figure 5 diagnostics-12-00799-f005:**
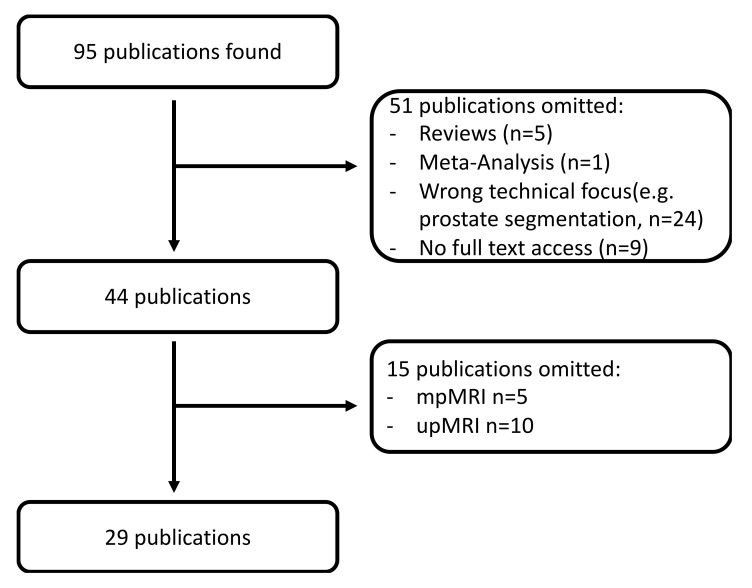
Literature selection work-flow. ML–machine-learning. DL–deep learning. up–uniarametric. bp–biparametric. mp–multiparametric.

**Figure 6 diagnostics-12-00799-f006:**
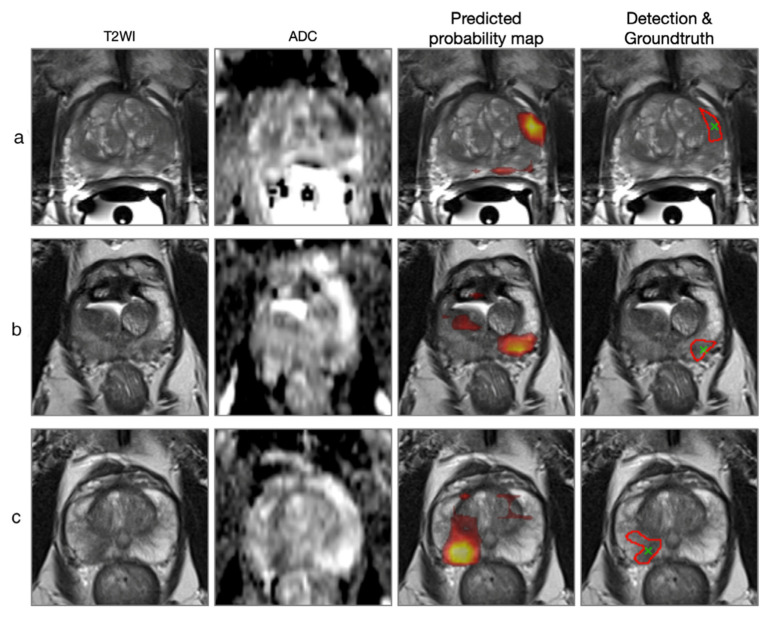
“Examples of lesion detection. The left two columns show the input T2WI and ADC map, respectively. The right two columns show the FocalNet-predicted lesion probability map and detection points (green crosses) with reference lesion annotation (red contours), respectively. (**a**) Patient at age 66, with a prostate cancer (PCa) lesion at left anterior peripheral zone with Gleason Group 5 (Gleason Score 4 + 5). (**b**) Patient at age 68, with a PCa lesion at left posterolateral peripheral zone with Gleason Group 2 (Gleason Score 3 + 4). (**c**) Patient at age 69, with a PCa lesion at right posterolateral peripheral zone with Gleason Group 3 (Gleason Score 4 + 3). ADC = apparent diffusion coefficient; T2WI = T2-weighted imaging“(reprinted with permission from [72], Copyright 2021 John Wiley and Sons).

**Table 1 diagnostics-12-00799-t001:** List of include studies and relevant key information.

Reference	Year	ML	DL	Field Strength	Target	Number of Patients	Age	SS/SP/Accuracy	AUC	Sequences Used
Abdollahi H. et al. [40]	2019	1	0	1.5 T	Gleason score prediction	33	73 (51–82)		0.739	T2, ADC
Wu M. et al. [48]	2019	1	0	3 T	TZ PCA detection	44	68 ± 7	93.2%/98.4%	0.989 (LR)	T2, ADC
Varghese B. et al. [49]	2019	1	0	3 T	Grading prediction	68 53		86%/72%	0.71	T2, ADC,
Min X. et al. [50]	2019	1	0	3 T	ci/csPCA discrimination TZ and PZ	280		84.1%/72.7%	0.823	T2, ADC, b1500
Toivonen J. et al. [51]	2019	1	0	3 T	Gleason prediction TZ and PZ	62	65 (45–73)		0.88	T2, b0-b2000, T2mapping
Chen T. et al. [52]	2019	1	0	3 T	Tumor detection aggressiveness prediction TZ and PZ	182 199	73 (55–90)	98.6/99.2%/98.9% (noPCA vs. PCA) 100/98.25 8/99.1% (ci vs. csPCA)	0.999 (noPCA vs. PCA) 0.933 (ciPCA vs. csPCA)	T2, ADC
Xu M. et al. [53]	2019	1	0	3 T	Tumor detection	331	71 (46–94)		0.92 (Radiomics) 0.993 (R + clinical data)	T2, ADC, DWI
Zhong X. et al. [54]	2019	0	1	3 T	ci/cs PCA discrimination DL vs. PIRADS exp. radiologists	140		63.6%/80.6%/72.3% 86.4%/48.0%/86.4%	0.726 (DL) 0.711 (PIRADS v2)	
Yuan Y. et al. [55]	2019	0	1	3 T	ci/cs PCS discrimination (GS > 7)	132 112		–/–/86.9%		T2 ax and sag, ADC
Xu H. et al. [56]	2019	0	1	3 T	Detection of PIRADS ≥ 3 lesions	346		–/–/93.0%	0.950	T2, ADC, high b-value
Schelb P. et al. [57]	2019	0	1	3 T	DL and radiologist for lesion (PIRADS ≥ 3 and 4) detection and segmentation	250 62	64 (58–71) 64 (60–69)	98/17% Rad, PIRADS ≥ 3 84/48% Rad, PRIADS ≥ 4 99/25%, DL, PIRADS ≥ 3 83/55%, DL, PIRADS ≥ 4		T2, ADC, DWI
Montoya Perez I. et al. [58]	2020	1	0	3 T	Detection of csPCA with bpMRI, RNA and clinical data	80	65 ± 7.1		0.92	T2, DWI
Hou Y. et al. [59].	2020	1	0	3 T	csPCA in PIRADS 3 identification in TZ and PZ	263	66.8 ± 11.4		0.89	T2, ADC, b1500
Mehralivand S. et al. [60]	2020	1	0	3 T	Detection csPCA in TZ and PZ	236		50.8%/–/– (TZ, MRI) 61.8%/–/– (TZ, DL)	0.749 (MRI) 0.775 (DL)	T2, b1500
Gong L et al. [61]	2020	1	0	3 T	ci/cs PCA discrimination	326 163		73.8%/65.8%/69.9%	0.788	T2, ADC, b800
Bleker J. et al. [62]	2020	1	0	3 T	ci/cs PCA discrimination in PZ	206	66 (48–83)		0.870 (mpMRI) 0.816 (bpMRI)	T2, ADC, DWI, (DCE)
Zong W. et al. [63]	2020	0	1	3 T	CNN optimization	367		100/92%	0.840	T2, ADC, b0
Sanford T. et al. [64]	2020	0	1	3 T	Automated PIRADS classification compared to radiologist	687	67 (46–89)			T2, ADC, high b-value
Brunese L. et al. [65]	2020	1	1	1.5 T	Gleason score prediction	52		–/–/98%		T2, DCE
Chen Y. et al. [66]	2020	0	1	3 T	Prostate and cancer segmentation	136	68 (49–62)	75.1/99.9%		T2, ADC, b1200
Winkel D.J. et al. [67]	2020	0	1	3 T	bpMRI PCA Screening	49	58 (45–75)	87/50%		T2, ADC, b2000
Arif M. et al. [68]	2020	0	1	3 T	Detection of csPCA in AS	292	68 (62–72)	92/76%	0.89	T2, ADC, b800
He D. et al. [69]	2021	1	0	3 T	Tumor detection Prediction ECE Prediction PSM	459	65 (30–89)		0.863 0.905 (integrated model)	T2, ADC
Vente C. et al. [70]	2021	0	1	3 T	csPCA detection and grading	99 63				T2, ADC
Chen J. et al. [71]	2021	0	1	3 T	csPCA detection and grading	25		89.6/90.2%/92.1%	0.964	T2, T1
Cao R. et al. [72]	2021	0	1	3 T	PCA detection and grading	126 427	62.4 ± 6.4 61.1 ± 7.1	98/17% PIRADS, ≥3 85/58% PIRADS, ≥4 100/17% Unet ≥ 3 83/58% Unet ≥ 4		T2, ADC
Hou Y. et al. [73]	2021	0	1	3 T	ECE prediction	590 150 103	69.2 (42–86) 69.2 (48–83) 70.2 (52–87)		0.857 0.728	T2, ADC, b1500
Yan Y. et al. [74]	2021	1	1	3 T	BCR prediction	485	69.8		0.802 (C-index)	T2
Schelb P. et al. [75]	2021	0	1	3 T	csPCA detection and grading	284	64 (IQR 61–72)	98/17% PIRADS, ≥3 85/55% PIRADS, ≥4 99/24% Unet ≥ 3 83/55% Unet ≥ 4		T2, ADC, b1500

**Table 2 diagnostics-12-00799-t002:** Display of study results comparing human and AI-based performance.

Reference	Year	ML	DL	Metric	Human Radiologist	AI-Approach
Chen T. et al. [52]	2019	1	0	AUC	0.867	0.999
Schelb P. et al. [57]	2019	0	1	Sensitivity/Specificity	98/17% PIRADS ≥ 3 84/48% PRIADS ≥ 4	99/25% PIRADS ≥ 3 83/55% PIRADS ≥ 4
Mehralivand S. et al. [60]	2020	1	0	AUC Sensitivity	0.816 89.6%	0.780 87.9%
Sanford T. et al. [64]	2020	0	1	Cancer detection rates	53% PIRADS 3 61% PRIADS 4 92% PIRADS 5	57%, PIRADS 3 60%, PIRADS 4 89% PIRADS 5
Cao R. et al. [72]	2021	0	1	Sensitivity/Specificity	98/17% PIRADS, ≥3 85/58% PIRADS, ≥4	100/17% PIRADS, ≥3 83/58% PIRADS, ≥4
Schelb P. et al. [75]	2021	0	1	Sensitivity/Specificity	98/17% PIRADS, ≥3 85/55% PIRADS, ≥4	99/24% PIRADS, ≥3 83/55% PIRADS, ≥4

## Data Availability

All data can be found in the original publications as listed above.

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
