# Peer review of "Current Value of Biparametric Prostate MRI with Machine-Learning or Deep-Learning in the Detection, Grading, and Characterization of Prostate Cancer: A Systematic Review"

_diagnostics, 2022, doi:10.3390/diagnostics12040799_

Round 1
Reviewer 1 Report
General comment:
The manuscript entitled “Current Value of Biparametric Prostate MRI with Machine-Learning or Deep-Learning in the Detection, Grading and Characterization of Prostate Cancer: a narrative review” aims to elucidate the status of AI in prostate cancer imaging, focusing on the role of bpMRI.
Although the topic is interesting and consistent with other studies reported in the literature, I do not understand if this is a narrative review or a systematic review. The section “materials and methods”, the PRISMA-like diagram and the findings reported, are more suitable to a systematic review while paragraphs and introduction are built as a narrative article. This is the main concern regarding you work. It makes the manuscript chaotic, fuzzy and not understandable. Choose which kind of manuscript you want and cull the excess.
- Major issue
INTRODUCTION
- 25-26: I would add a brief report on main risk factors of prostate cancer. Please see: https://doi.org/10.3390/diagnostics11050908 and https://doi.org/10.1016/j.euf.2018.02.007
MATERIALS AND METHODS
- As reported in the general comment, I do not fully understand: if this is a narrative review, you do not need a “standardized” materials and methods section. However, if you add number of studies involved/excluded, PRISMA diagram and search queries, this make your article more “systematic-like” oriented. For example, 203 to 213 is more suitable to a systematic review.
- Moreover, paragraphs 2.1 and 2.2 are not clear and should be revised.
- In machine learning you should add the issue regarding ROI
- 217-236: this section should be better placed in the discussion, together with the limitations.
DISCUSSION
- Limitations should be moved from conclusions to this section.
FIGURES
- Although you obtained the permission, I would limit the number of non-original figures.
- Minor issue
Check grammar and typos. References should be added at the end of sentences.
ABSTRACT
- 17-18: Not clear, please revise.
INTRODUCTION
- 31: add references
- 32-35: Highlight the difference in sensitivity and specificity of MRI compared to US
- 49: if you report BIRADS, you should at least add a reference.
- 56: regarding this, see: https://doi.org/10.3390/ijms22189971 and https://doi.org/10.1007/s00066-020-01679-9
RESULTS
- 257: this paragraph is too long and chaotic
DISCUSSION
- 446: references needed at the end of sentence. There are similar sentence requiring references along the discussion as well.
- 480: avoid the repetition of sentences from other studies, even as a citation.
Author Response
Reviewer 1
General comment:
The manuscript entitled “Current Value of Biparametric Prostate MRI with Machine-Learning or Deep-Learning in the Detection, Grading and Characterization of Prostate Cancer: a narrative review” aims to elucidate the status of AI in prostate cancer imaging, focusing on the role of bpMRI.
Although the topic is interesting and consistent with other studies reported in the literature, I do not understand if this is a narrative review or a systematic review. The section “materials and methods”, the PRISMA-like diagram and the findings reported, are more suitable to a systematic review while paragraphs and introduction are built as a narrative article. This is the main concern regarding you work. It makes the manuscript chaotic, fuzzy and not understandable. Choose which kind of manuscript you want and cull the excess.
we agree. The title was changed accordingly, the manuscript adapted.
- Major issue
INTRODUCTION
- 25-26: I would add a brief report on main risk factors of prostate cancer. Please see: https://doi.org/10.3390/diagnostics11050908 and https://doi.org/10.1016/j.euf.2018.02.007 we agree. Changes undertaken.
MATERIALS AND METHODS
- As reported in the general comment, I do not fully understand: if this is a narrative review, you do not need a “standardized” materials and methods section. However, if you add number of studies involved/excluded, PRISMA diagram and search queries, this make your article more “systematic-like” oriented. For example, 203 to 213 is more suitable to a systematic review.--> we agree, the manuscript was changes accordingly.
- Moreover, paragraphs 2.1 and 2.2 are not clear and should be revised. we agree, the sections were restructured and shortened
- In machine learning you should add the issue regarding ROI we agree, the document was adapted accordingly.
- 217-236: this section should be better placed in the discussion, together with the limitations. we do not agree and left this section unchanged. If it is the editors wish to move it back to the discussion, then we will do so.
DISCUSSION
- Limitations should be moved from conclusions to this section. we do not agree and left this section unchanged. If it is the editors wish to move it back to the discussion, then we will do so.
FIGURES
- Although you obtained the permission, I would limit the number of non-original figures. we do not agree. The figures included were chosen on purpose and are important to underline the key messages of this manuscript. If it is the wish of the editor to do so, please advise us on which figure should be removed.
- Minor issue
Check grammar and typos. References should be added at the end of sentences. we agree. Changes performed.
ABSTRACT
- 17-18: Not clear, please revise. we agree, the manuscript was corrected.
INTRODUCTION
- 31: add references we agree, the manuscript was corrected
- 32-35: Highlight the difference in sensitivity and specificity of MRI compared to US we agree, the manuscript was corrected
- 49: if you report BIRADS, you should at least add a reference. we agree, the manuscript was corrected
- 56: regarding this, see: https://doi.org/10.3390/ijms22189971 and https://doi.org/10.1007/s00066-020-01679-9 we disagree. There is no understandable need to have these references introduced here. If it is the editors whish to have them here, we will do so.
RESULTS
- 257: this paragraph is too long and chaotic à we agree, the manuscript was corrected
DISCUSSION
- 446: references needed at the end of sentence. There are similar sentence requiring references along the discussion as well. we agree, the manuscript was corrected
- 480: avoid the repetition of sentences from other studies, even as a citation. we agree, the manuscript was corrected
Reviewer 2 Report
The authors of this paper try to summarize the evidence about a very hot topic in prostate imaging. Despite very interesting, the review should be improved and restructured before been able to be published.
Some comments are the following:
ABSTRACT:
Some summaries information concluded from the review should be shared within the abstract.
Material and methods.
The review should be better structured. In material and methods at least a brief description of how the authors perform the literature search and which questions they have tried to answer should be explained.
Probably all of the MRI description should be included in the introduction section, and material and methods specify just the literature review and the specific questions answered.
Probably;
- comparison between bi and multiparametric (in this scenario there are a lot more literature than the summarize in the article, and future trials such as PRIME TRIAL should be mentioned).
- AI in MRI
Please review the writing. there are some incomplete sentences such as line 319 or not changed 532-536.
Some studies interpretation seems wrong;.i.e. IMPROD trial conclusion literary state “the use of radiomics and kallikreins failed to outperform PI-RADSv2.1/IMPROD bpMRI Likert and their combination did not lead to further performance gains”. It does not look the same as “From the prospective IMPROD trial a sub analysis examined if the addition of clinical
data and RNA expression profiles of genes associated with prostate cancer increased the accuracy for detection of csPCA [46]. In this study the bpMRI based data yielded the high- est AUC 0.92. Adding RNA-based data or clinical data did neither improve the results nor yield better results by itself”.
Globally the analysis and evidence provided shows not clear advantage for AI in MRI, however the authors specifically focus in a section with PIRADS 3 in which this kind of technology could gain more value. Could the author extract information from the other papers about specifically this scenario? It´d clearly improve the interest of the paper.
A small section of limitations should be included focus on limitations of the field studied and of the review itself.
Author Response
Reviewer 2
ABSTRACT:
Some summaries information concluded from the review should be shared within the abstract. we agree, the abstract was amended.
Material and methods.
The review should be better structured. In material and methods at least a brief description of how the authors perform the literature search and which questions they have tried to answer should be explained. we agree, the manuscript was amended.
Probably all of the MRI description should be included in the introduction section, and material and methods specify just the literature review and the specific questions answered. we agree, the manuscript was rearranged.
Probably;
- comparison between bi and multiparametric (in this scenario there are a lot more literature than the summarize in the article, and future trials such as PRIME TRIAL should be mentioned).
please provide more information on the PRIME TRIAL. The only somewhat pertinent trial I found focused on radiotherapy methods which was not the focus of this review (Randomised controlled trial of Prostate Radiotherapy In high risk and node positive disease comparing Moderate and Extreme hypofractionation [PRIME Trial]).
- AI in MRI
Please review the writing. there are some incomplete sentences such as line 319 or not changed 532-536. we agree, the sentences were adapted.
Some studies interpretation seems wrong;.i.e. IMPROD trial conclusion literary state “the use of radiomics and kallikreins failed to outperform PI-RADSv2.1/IMPROD bpMRI Likert and their combination did not lead to further performance gains”. It does not look the same as “From the prospective IMPROD trial a sub analysis examined if the addition of clinical
data and RNA expression profiles of genes associated with prostate cancer increased the accuracy for detection of csPCA [46]. In this study the bpMRI based data yielded the high- est AUC 0.92. Adding RNA-based data or clinical data did neither improve the results nor yield better results by itself”.
we agree, the manuscript was changed accordingly.
Globally the analysis and evidence provided shows not clear advantage for AI in MRI, however the authors specifically focus in a section with PIRADS 3 in which this kind of technology could gain more value. Could the author extract information from the other papers about specifically this scenario? It´d clearly improve the interest of the paper. unfortunately no other papers provided separate PIRADS 3 data.
A small section of limitations should be included focus on limitations of the field studied and of the review itself. we agree, the manuscript was amended accordingly.
Reviewer 3 Report
The manuscript is well structured and detailed with expressing the procedural steps. It reads well with easy to track format. However, the authors' attention is needed for the following issues,
- Please keep the consistency for using the abbreviation, MR and MRI are not stand for the same methodologies.
- Please abbreviate the phrases at their first use. There are multiple occasions for this issue.
- Please revise the article for missing references, e.g. lines 31, 35, 64, 100, 117. When you're mentioning some approaches as the gold standard, it better should cite the appropriate articles.
- Please check for unnecessary use of articles and missing punctuation marks.
- Please check your article for the typos, e.g. lines 276, 355, 356 and etc.
- As citing the references by using the first author's name, you only need to use the surname, not his/her first name.
- Also, the page numbers are not consistent. Please check it.
- Please revise the statements starting at lines 78, 117, and 225.
- Please correct the data availability statement.
- Please check reference 68 for completeness.
- Please check figure 6.
Author Response
Reviewer 3
- Please keep the consistency for using the abbreviation, MR and MRI are not stand for the same methodologies. we agree. The manuscript was checked and corrected where appropriate.
- Please abbreviate the phrases at their first use. There are multiple occasions for this issue.
- Please revise the article for missing references, e.g. lines 31, 35, 64, 100, 117. When you're mentioning some approaches as the gold standard, it better should cite the appropriate articles.
- Please check for unnecessary use of articles and missing punctuation marks. we agree. The manuscript was checked and corrected where appropriate.
- Please check your article for the typos, e.g. lines 276, 355, 356 and etc. à we agree. The manuscript was checked and corrected where appropriate.
- As citing the references by using the first author's name, you only need to use the surname, not his/her first name. we agree. The manuscript was checked and corrected where appropriate.
- Also, the page numbers are not consistent. Please check it. we agree. The manuscript was checked and corrected where appropriate.
- Please revise the statements starting at lines 78, 117, and 225. we agree. The manuscript was checked and corrected where appropriate.
- Please correct the data availability statement. we checked and corrected.
- Please check reference 68 for completeness. we agree. The manuscript was checked and corrected where appropriate.
- Please check figure 6. we checked and corrected.
Reviewer 4 Report
This manuscript gave a review on Biparametric Prostate MRI with machine learning techniques to help detect and grade prostate cancer. 29 appropriate studies were selected and analyzed. The promising evaluation results in those studies demonstrate a trend to apply AI-postprocessing techniques and a broader clinical application in the future.
Minor comments:
1. Please double check the all the figure in the manuscript, e.g.,
In Figure 6, what does “no title” mean?
Author Response
Minor comments:
- Please double check the all the figure in the manuscript, e.g.,
In Figure 6, what does “no title” mean? typo. It was corrected.
Round 2
Reviewer 1 Report
Systematic, not systemic.
Author Response
Thank your for your advise. The error has been corrected.
Reviewer 2 Report
The authors did a good job trying to improve the review according to the comments, however there are still thing to improve.
Please rewrite and change systemic to systematic… However, to clearly denominate a paper as a systematic review, this should follow a specific PRISMA guidelines. (doi:1136/bmj.n160). Please share a table double checking the authors follow the items required.
The following sentence: “PCA aggressiveness can be linked to specific 32 genes such as BRCA and behaviour such as smoking [4,5]”, is not related with the specific focus of the article and from my point of view, there are several more important factors linked with prostate cancer aggressiveness, so this sentence should be withdrawal.
In the bi and multiaparametric section my proposal was to explain deeper the evidence. Prime trial was just and explample of a planned trial to give better evidence about the comparison (Prostate Imaging Using MRI +/- Contrast Enhancement” (NCT04571840 ).
Author Response
Thank you for your valuable advise.
Please rewrite and change systemic to systematic… However, to clearly denominate a paper as a systematic review, this should follow a specific PRISMA guidelines. (doi:1136/bmj.n160). Please share a table double checking the authors follow the items required. --> we agree, a table was added to the appendix
The following sentence: “PCA aggressiveness can be linked to specific 32 genes such as BRCA and behaviour such as smoking [4,5]”, is not related with the specific focus of the article and from my point of view, there are several more important factors linked with prostate cancer aggressiveness, so this sentence should be withdrawal. --> this section was introduced as it was requested by reviewer 1. It's is at the editor's discretion whether it can stay or should be removed.
In the bi and multiaparametric section my proposal was to explain deeper the evidence. Prime trial was just and explample of a planned trial to give better evidence about the comparison (Prostate Imaging Using MRI +/- Contrast Enhancement” (NCT04571840 ). --> we agree, the section was expanded.
Round 3
Reviewer 2 Report
The authors have followed the previous recommendations.